# Robust Automatic Segmentation of Inflamed Appendix from Ultrasonography with Double-Layered Outlier Rejection Fuzzy C-Means Clustering

Kwang Baek Kim [1,*], Doo Heon Song [2] and Hyun Jun Park [3]

1   Department of Artificial Intelligence, Silla University, Busan 46958, Korea
2   Department of Computer Games, Yong-In Art & Science University, Yongin 17145, Korea; dsong@yasu.ac.kr
3   Division of Software Convergence, Cheongju University, Cheongju 28503, Korea; hyunjun@cju.ac.kr
*   Correspondence: gbkim@silla.ac.kr

**Abstract:** Accurate diagnosis of acute appendicitis from abdominal ultrasound is a challenging task, since traditional sonographic diagnostic criteria for appendicitis, such as diameter, compressibility, and wall thickness, rely on complete identification or visualization of the appendix and the diagnosis is frequently operator subjective. In this paper, we propose a robust automatic segmentation method for inflamed appendix identification to mitigate abovementioned difficulties. We use outlier rejection fuzzy c-means clustering (FCM) algorithm within a double-layered learning structure to extract the target inflamed appendix area. The proposed method extracts the target appendix in 98 cases out of 100 test images, which is far better than traditional FCM, standard outlier FCM, and double-layered learning with FCM in correct extraction rate. Furthermore, we investigate the outlier rejection effect and double layered learning effect by comparing our proposed method with standard double-layered FCM and the standard outlier-rejection FCM. In this comparison, the proposed method exhibits robust segmentation results in accuracy, precision, and recall by 2.5~5.6% over two standard methods in quality with human pathologists' marking as the ground truth.

**Keywords:** inflamed appendix; appendicitis; segmentation; outlier rejection; fuzzy c-means; pixel clustering; double-layered learning





## 1. Introduction

The appendix is a finger-like, blind-ended tube connected to the cecum. Normal vermiform appendix is less than 6mm in diameter and has a thin wall. Appendicitis refers to inflammation of the appendix and it is a well-known, common, and painful abdominal surgical emergency [1]. It can lead to perforation of the organ if left untreated; thus, once diagnosed as acute appendicitis, prompt surgical treatment is required. Depending on the individual patient's physical condition and medical history, careful treatment, such as the time of operation, selection of an incision method, performing drainage or not, treatment of wounds, and administration of antibiotics, should be planned.

An obstruction, or blockage, of the appendiceal lumen causes appendicitis. The most common cause of clogged lumen is feces, but there are other causes such as lymphatic tissue proliferation, debris from previous barium tests, vegetables, fruit seeds, or parasites [2]. When the lumen of the appendix is blocked, closed loop obstruction occurs and secretion of mucus continues into the lumen of the appendix, causing the lumen of the appendix to expand, thereby continuously increasing the pressure in the lumen of the appendix. This lumen dilatation increases the peristalsis of the appendix, causing cramping pain in addition to visceral pain. The continuous increase in internal pressure leads to venous stasis of the appendix wall, which leads to bacterial invasion and inflammation of the suppurative appendicitis as the mucosal defenses are impaired. If it progresses further and tissue ischemia is caused by arterial supply disturbance, it becomes the stage of

gangrene appendicitis, in which necrosis and serous exudate occur throughout the wall of the appendix.

The diagnosis of acute appendicitis is a challenging task. Detection of appendiceal neoplasms at pre-operative imaging is important because it may change the surgical approach and prevent additional surgery [3]. However, more than half will have different clinical and radiological manifestations [4] and there exist non-negligible amount of age and gender related false positive cases, such as pelvic inflammatory disease and other gynecologic conditions for childbearing age women [5,6]. Such misdiagnosis or delayed diagnosis may increase the incidence of perforation, peritonitis, and negative laparotomy, which can affect the morbidity and mortality of patients [7]. Thus, it is important to have a quick and accurate acute appendicitis diagnosing methodology for patients appealing acute abdominal pain.

Among possible diagnostic modalities, ultrasound examination is usually the first imaging test performed, especially for the pediatric and young adult populations, as well as patients in pregnancy who are very vulnerable to the radiation. However, if the first opinion of ultrasound examination is in indeterminate status with a strong clinical suspicion of appendicitis, a second ultrasound examination by well experienced staff or a Computed tomography (CT) scan or Magnetic resonance imaging (MRI) scan is necessary for reliable diagnosis [8,9].

Appendiceal ultrasound is most accurate when the appendix is clearly identified during the examination and relevant sonographic criteria is applied to determine the presence of appendiceal inflammation [10]. Traditional sonographic diagnostic criteria for appendicitis, such as diameter, compressibility, wall thickness, and hyperaemia, rely on complete identification or visualization of the appendix. However, according to the recent report, the sonographic appendix visualization rate can vary significantly [11]. It becomes more challenging when abdominal ultrasound could not correctly visualize the inflamed appendix [12]. Moreover, while the reported specificity of ultrasound is almost equal to CT, the reported sensitivity for diagnosis by ultrasound varies wider in the literature (67–100%) and showed high operator dependency [13]. A correct ultrasound technique and knowledge about imaging features of appendicitis improve the detection of inflamed appendix and therefore may help to make the diagnosis without performing unnecessary CT scans [14].

To avoid such operator subjectivity and unclear visualization of inflamed appendix from abdominal ultrasound, we need robust automatic segmentation tool to extract inflamed appendix. Simple edge detecting methods relying on histogram analysis with thresholds [15–17] are inaccurate when the brightness contrast of the image is not remarkably high. The reported performance to extract inflamed appendix were no better than 83.3% due to information loss during edge linking process.

To identify the target object from low contrasted ultrasonography, pixel clustering method is a viable alternative and has been proven to be effective in engineering and medical domains [18–22]. In this pixel-based approach, the automatic segmentation process is usually divided into two phases—brightness enhancement for noise reduction and quantization-based object formation. The basic predictor of locating inflamed appendix is the fascia line [23]. Fuzzy Adaptive resonance theory (ART) based quantization algorithm for inflamed appendix extraction [24] was immune to that brightness sensitivity and showed better extraction rate than edge detecting methods, but it was found that the method was weak when the shape of the inflamed appendix has a certain type. Self-Organizing Map (SOM) is another pixel clustering method with structural stability. Thus, the SOM based method [25] has shown better results but, due to its winner-takes-all strategy in the learning process, the fatal irreversible loss of intensity information could cause the failed extraction of appendix when the shape has long oval pattern. Fuzzy C-Means (FCM) based algorithm with semi-dynamic control of initialization process was almost perfect in extracting inflamed appendix if exists [26].

However, in terms of object identification, previous studies only reported if the software could locate the target if the inflamed appendix were given. Such previous works used 6mm single cutoff standard to extract the target according to the clinical experts' suggestions, but recent report suggests a ternary classification of appendix diameter (negative, equivocal, and positive) with cut-offs at 6 and 8 mm and showed better clinical classification result [11]. Thus, for practical applications, an accurate automatic localization of the appendix region is more important than finding inflamed appendix having a certain size.

Thus, in this paper, we propose a robust segmentation algorithm to locate the inflamed appendix from ultrasound that has two articulating points from previous studies.

1. We compare the location of target object with human expert's drawing from the same ultrasonographic images as tried in other medical tasks [19,27]. By doing so, we want our segmentation algorithm acts like human expert;
2. We propose Double-layered Outlier Rejection Fuzzy C-Means (DORFCM) as the quantization algorithm that mitigates the outlier sensitivity of FCM.

FCM based segmentation has shown many successful applications in many image analysis domains [27–31]; however, its sensitivity to noise and outliers have been criticized [28]. To mitigate such claimed deficiency, there are numerous modified versions of FCM such as having weighted features in consideration [32,33], using alternative distance measure [34], and outlier rejection [35]. In this paper, we adopt outlier rejection strategy in detecting inflamed appendix for the robustness of segmentation. Furthermore, we have double layered learning structure used in [36] for the stable learning. Thus, our proposed method in this paper is a hybrid FCM learning method with outlier rejection strategy and double layered learning structure to make the inflamed appendix detection to be more robust in quality.

## 2. Appendix Segmentation with Double-Layered Outlier Rejection FCM
### 2.1. Outlier Rejection

Based on fuzzy set theory, FCM [37] often generates better results in image segmentation than hard clustering due to its tolerance to ambiguity of object membership on the boundaries between several clusters. Like K-means, it computes the distance between pixels and cluster centers and its cost function is minimized when pixels close to the centroid of their clusters are assigned higher membership values than others. However, the intra-cluster variance and the inter-cluster variance are not optimized when there exist overlapping of regions or outliers [35].

Outlier Rejection FCM(ORFCM) proposed in [35] modified the membership function to overcome the outlier's sensitivity by changing the objective function of FCM as following.

$$\min ORFCM = \sum_{i=1}^{n} \sum_{j=1}^{k} u_{ij}^{m} \beta \|x_i - c_j\|^2 \tag{1}$$

where $X = \{x_i\}$, $i \in \{1, 2, \ldots, n\}$ denotes an image with $n$ pixels to be partitioned into $k$ clusters, where $2 \le k \le n$, and $c_j$ (for $j = 1, 2, \ldots, k$) is the $j$-th cluster. $U = (u_{ij})$ is a $k \times n$ fuzzy partition matrix, in which each element $u_{ij}$ indicates the membership degree of each pixel in the $j$-th cluster, $c_j$.

The exponent variable $\beta$ limits the partial distribution of the points among the two neighboring clusters rather than to all the clusters.

$$\beta = \frac{intensity \ in \ an \ cluster + 1}{256} \tag{2}$$

With this exponent operation of the Euclidean distance in its membership function, ORFCM showed its capability to neutralize the outlier's effect and construct better clusters than the original FCM. There exist several recent results in medical domain using this outlier rejection concept in fuzzy learning process successfully [38–40].

### 2.2. Double-Layered Pixel Clustering

Another well-known limitation of FCM is its tendency to fall into local maximum due to the gradient descent method it is based on. A recent study [36] overcome that limitation by making the FCM learning structure as double-layered as shown in Figure 1.

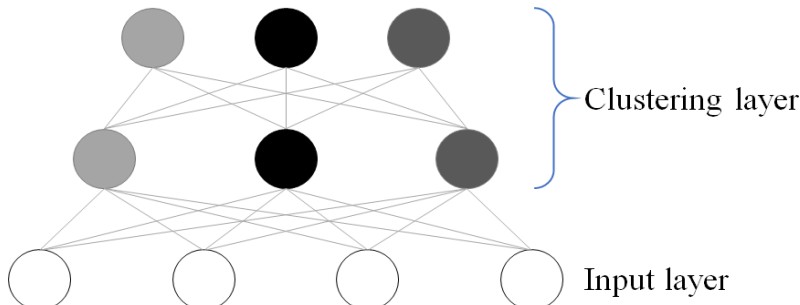

**Figure 1.** Double-layered learning structure.

The second layer takes outputs of the first layer as input. Subsequent parameters of the first and second layers of clustering are not necessarily the same. In [36], they used conventional FCM in both layers but with different parameter initializations and the number of clusters.

### 2.3. Double-Layered Outlier Rejection FCM (DORFCM)

In this paper, we propose double-layered outlier rejection FCM (DORFCM) incorporating both concepts. Two clustering layers use ORFCM as their clustering engine but with different learning parameters. In the first layer, the number of clusters for ORFCM is initialized by the histogram analysis. In the second layer, the intensity value is controlled in that outlier pixels are removed from the previous learning process. Thus, the optimizing process of the second layer has different parameters to avoid misguidance from the first layer learning.

The algorithm of DORFCM can be summarized as follows.

#### 2.3.1. Layer-1 Learning (Step 1 to Step 4)

**Step 1:** Initialize $c$ ($2 \leq c < n$) as the number of clusters, and exponential weight $m$ ($1 \leq m < \infty$), and initialize the error threshold ($\varepsilon$) for terminating condition of the first layer learning and the membership degree $U(0)$, repetition parameter $r = 0$.

**Step 2:** Compute the value of central vector $v_{ij}$ as shown in Equation (3) for $\{v_i \mid j = 1, 2, \ldots, c\}$.

$$v_{ij}^{(g)} = \frac{\sum_{k=1}^{n}(U_{ik})^m X_{kj}}{\sum_{k=1}^{n}(U_{ik})^m} \tag{3}$$

where $X$ is the input pattern, $i$ is the cluster index, and $j$ is the pattern node index. $k$ is the pattern index, $n$ is the number of patterns, and $U$ is the membership function.

**Step 3:** Define the ORFCM cost function as Equations (4) and (5) with $\beta$ as Equation (2).

$$d_{ik} = \left[\sum_{j=1}^{l}\left(x_{kj} - v_{ij}\right)^2\right]^{\frac{1}{2}} \tag{4}$$

$$u_{ik}^{(g+1)} = \frac{1}{\sum_{j=1}^{c}\left[\frac{\beta \times d_{ik}^r}{\beta \times d_{jk}^r}\right]^2} \text{ for } I_k = \varnothing \tag{5}$$

where the distance $d_{ik}$ is defined as the Euclidean distance between the $k$-th pattern $x_k$ and $v_i$, the central vector of the $i$-th cluster, and $u_{ik}$ is the membership degree of $x_k$ among patterns in the $i$-th cluster.

**Step 4:** Compute the difference ($U_{ik}(r+1) - U_{ik}(r)$) between the new membership and the previous membership degree at the time of $r$. If the difference is less than the error threshold ($\varepsilon$), then the algorithm goes to Step 5, otherwise go to Step 2.

### 2.3.2. Layer-2 Learning (Step 5 to Step 7)

**Step 5:** With output of the first layer ORFCM as the input of the second layer, compute the value of central vector $w_{si}$ as shown in Equation (6) for $\{w_{si} \mid s = 1, 2, \ldots, cc\}$ where $cc$ is the number of clusters for the second layer.

$$w_{si}^{(t)} = \frac{\sum_{s=1}^{cc}(q_{si})^m u_{ik}}{\sum_{i=1}^{cc}(q_{si})^m} \tag{6}$$

**Step 6:** Compute the membership degree of the clusters $q_{si}$ in the second layer with distance function $d_{sk}$ as Equations (7) and (8) respectively. $p_s$ is the intensity in a cluster of layer-2.

$$q_{si}^{(t+1)} = \frac{1}{\sum_{i=1}^{cc}\left(\frac{\beta\|q_s - u_{ik}(t)\|}{\beta\|q_i - u_{ik}(t)\|}\right)^{\frac{2}{(m-1)}}} \tag{7}$$

$$\beta = \frac{p_s + 0.01}{256} \tag{8}$$

**Step 7:** Compute the difference between the new membership and the previous membership degree. If the difference is less than the final error threshold ($\varepsilon'$), then the algorithm terminates otherwise go to Step 5.

During the second layer learning, the value of $\beta$ is computed differently from the first layer learning since values of numerator and denominator are the results of first layer clustering, they should not exceed 1.

The flow of the proposed method can be illustrated as shown in Figure 2 and the Pascal-like pseudo code is given as Algorithm 1.

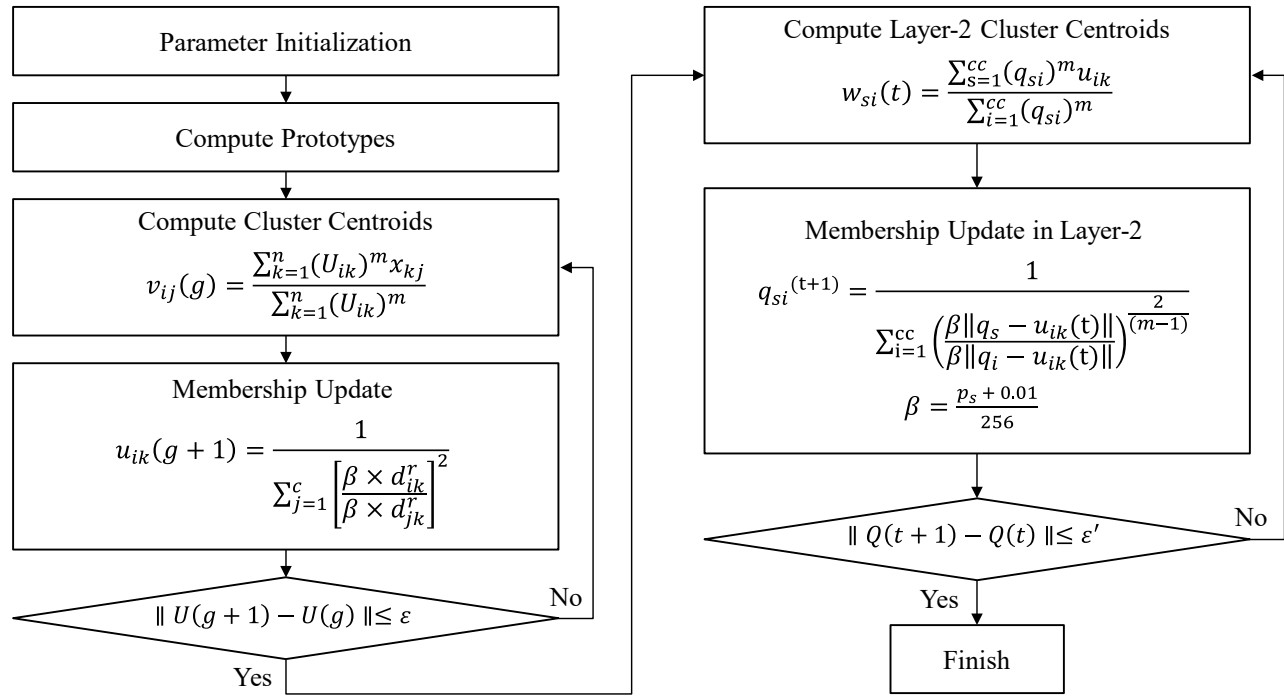

**Figure 2.** DORFCM flow chart.

---

**Algorithm 1.** The proposed method algorithm

---

**Procedure** DORFCM (x) **returns** prototypes and partition matrix
**Input** data x = {$x_1, x_2, \ldots, x_k$}
**Local** fuzzification parameter: m
    threshold: $\varepsilon$, norm: $\| \|$
    INITIALIZE-PARTITION-MATRIX
    g $\leftarrow$ 0, t $\leftarrow$ 0
    **repeat**
        **for** i = 1:c **do**
$$v_{ij}(g) \leftarrow \frac{\sum_{k=1}^{n} (U_{ik})^m x_{kj}}{\sum_{k=1}^{n} (U_{ik})^m} \text{ compute prototypes}$$
           **for** i = 1:c **do**
               **for** k = 1:n **do**
$$u_{ik}(g+1) \leftarrow \frac{1}{\sum_{j=1}^{c} \left[ \frac{\beta \times d_{ik}^r}{\beta \times d_{jk}^r} \right]^2} \text{ for } I_k = \varnothing \text{ update partition matrix}$$
          g $\leftarrow$ g + 1
    **until** $\| U(g+1) - U(g) \| \leq \varepsilon$
    **repeat**
        **for** s = 1:cc **do**
$$w_{si}(t) \leftarrow \frac{\sum_{s=1}^{cc} (q_{si})^m u_{ik}}{\sum_{i=1}^{cc} (q_{si})^m} \text{ Layer compute prototypes}$$
           **for** s = 1:cc **do**
               **for** i = 1:c **do**
$$q_{si}(t+1) \leftarrow \frac{1}{\sum_{i=1}^{cc} \left( \frac{\beta \| q_s - u_{ik}(t) \|}{\beta \| q_i - u_{ik}(t) \|} \right)^{\frac{2}{(m-1)}}} \text{ update Layer-2 partition matrix}$$
$$\beta = \frac{p_s + 0.01}{256} \text{ where } p_s \text{ is the intensity in a cluster of layer-2.}$$
        t $\leftarrow$ t + 1
    **until** $\| Q(t+1) - Q(t) \| \leq \varepsilon'$
    **return** w, q

---

## 3. Experiment and Analysis

The proposed method is implemented with C# under Microsoft Visual Studio 2019 on an IBM-compatible PC with Intel(R) Core (TM) i5-8250U 1.8GHz CPU and 8GB RAM. The experiment uses 100 DICOM format ultrasound images containing the inflamed appendix all from Gupo Sungsim Hospital, Busan, Korea, and there was no appendiclolith case in this experiment. Abdominal ultrasonography is performed by Philips iU22 using 3~5 MHz transducer. If the subject complains of pain on a specific body part, 7.5 Hz high frequency transducer is used to examine that part.

To verify the performance of the proposed method, we also implemented the original FCM, standard ORFCM [35] and Double-layered FCM(DFCM) used in [36]. All four methods have the same preprocessing and post-processing (object formation) steps, thus, any difference in performance is solely due to the algorithmic power.

The first performance index is the extraction rate where two human pathologists from the associated hospital confirms if each method successfully finds inflamed appendix from the input image. The result is summarized in Table 1.

**Table 1.** Extraction rate of inflamed appendix.

| Method | Successful Images/Total Images |
|:---:|:---:|
| FCM | 83/100 |
| ORFCM | 88/100 |
| DFCM | 91/100 |
| DORFCM | 98/100 |

Although verified as "success", there can be qualitative differences between segmentation methods. Figure 3 shows an easy task that all algorithms successfully find inflamed appendix object.

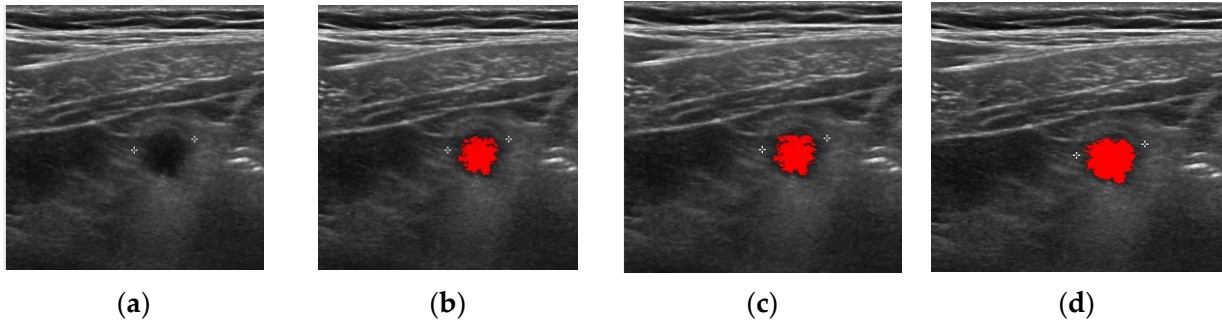

**Figure 3.** Extraction of inflamed appendix by methods. (**a**) Input, (**b**) FCM, (**c**) DFCM, and (**d**) DORFCM.

FCM is the weakest among four methods and Figure 4 shows the case that only FCM failed to locate inflamed appendix. FCM tends to underestimate the target area. Double-layered learning structure with FCM (Figure 4c) shows better quality than single FCM learning.

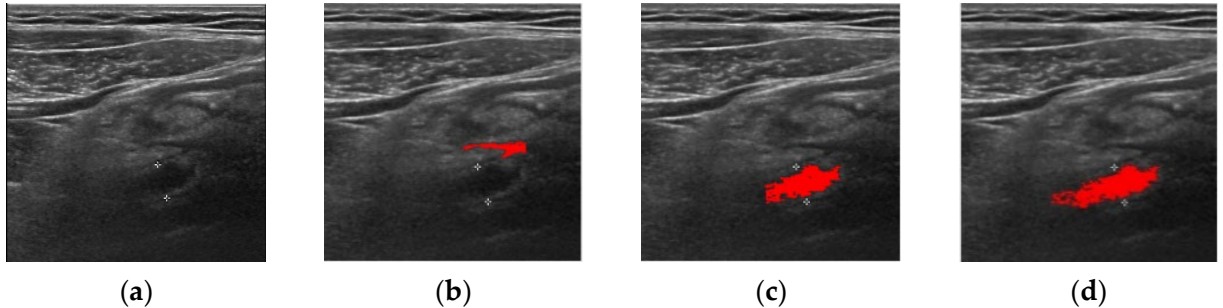

**Figure 4.** FCM failed to locate the right area. (**a**) Input, (**b**) FCM, (**c**) DFCM, and (**d**) DORFCM.

DFCM showed better performance than FCM to avoid local maximum in clustering. However, the outlier effect can mislead other methods, but DORFCM as one is verified in Figure 5. Two methods without outlier treatment failed to locate the right region and DFCM looks like it suffered by false positive and tends to overestimate when it fails.

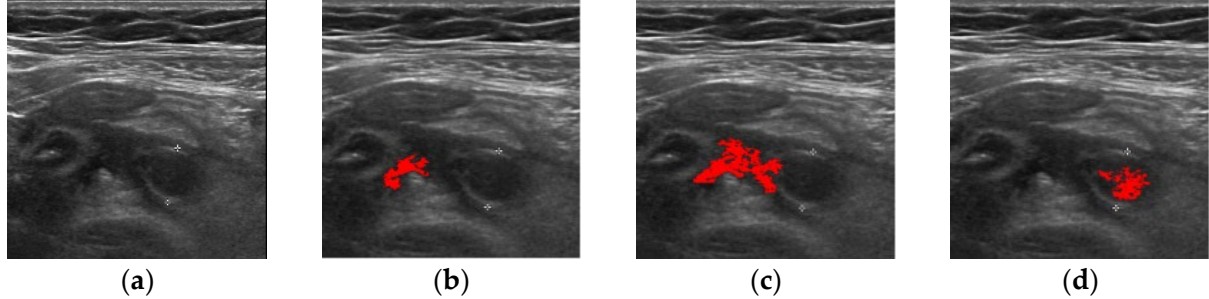

**Figure 5.** Only DORFCM succeeds. (**a**) Input, (**b**) FCM, (**c**) DFCM, and (**d**) DORFCM.

The standard ORFCM showed better result than FCM but worse than double-layered algorithms in inflamed appendix detection accuracy as shown in Table 1, but it searches different areas from FCM with outlier rejection strategy. However, our DORFCM showed better quality than single ORFCM and standard DFCM in cluster quality as shown in Figure 6.

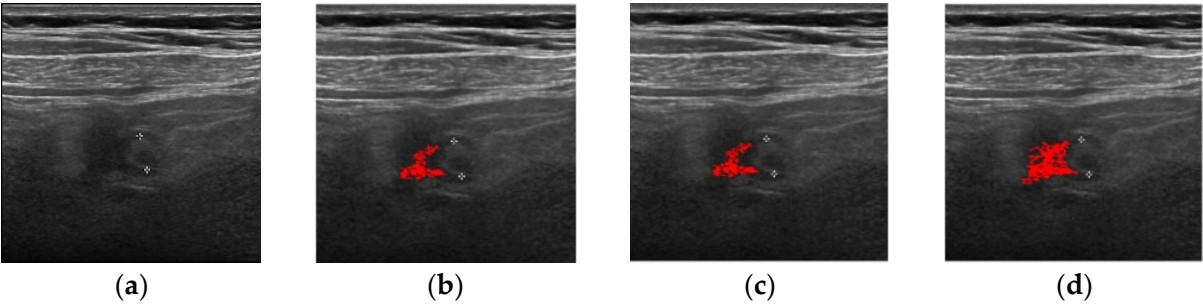

**Figure 6.** Comparison of DFCM, ORFCM, and DORFCM. (**a**) Input, (**b**) DFCM, (**c**) ORFCM, and (**d**) DORFCM.

Thus, we provide the second performance index is to compare the qualitative performance between DFCM, ORFCM, and DORFCM with respect to the human expert's area marking as the ground truth. True positive (TP) is the set of pixels that both human and the software agrees that they belong to the inflamed appendix, false positive (FP) is the set of pixels that only the software thinks they belong to the inflamed appendix, and false negative (FN) is the set of pixels that only human expert classifies as the inflamed appendix.

The performance index is as following.

Precision = TP/(TP + FP)
Recall or Sensitivity = TP/(TP + FN)
Accuracy = (TP + TN)/all area

Table 2 summarizes the performance comparison between DFCM and DORFCM among 91 cases that both methods locate the target object successfully. As one can see, the outlier rejection effect is significant.

**Table 2.** Outlier effect performance comparison (91 cases).

|  | Accuracy | Precision | Recall |
|---|---|---|---|
| DORFCM | 84.20% | 83.40% | 85.80% |
| DFCM | 80.80% | 80.90% | 80.20% |

Likewise, Table 3 summarizes the performance comparison between ORFCM and DORFCM among 88 cases that both methods locate the target object successfully. As one can see, the outlier rejection effect is significant.

**Table 3.** Double layered learning effect comparison (88 cases).

|  | Accuracy | Precision | Recall |
|---|---|---|---|
| DORFCM | 84.82% | 83.78% | 86.04% |
| ORFCM | 81.23% | 78.56% | 80.53% |

The effect of double-layered learning with outlier rejection strategy is shown in Figure 7. The standard ORFCM is implemented as the lower layer (layer-1) of our proposed DORFCM. With the input image Figure 7a, the number of clusters is initialized as 4 by analyzing the histogram as shown in Figure 7b and the segmentation result is shown as Figure 7c. For DORFCM, after lower layer learning, we can remove noise pixels that have lower membership rate and regulating intensities of similar pixels; thus, the histogram of layer-2 (higher layer) looks like Figure 7d thus DORFCM can avoid misguidance of ORFCM that is sensitive to outlier pixels. In the results, DORFCM shows better quantization as shown in Figure 7e.

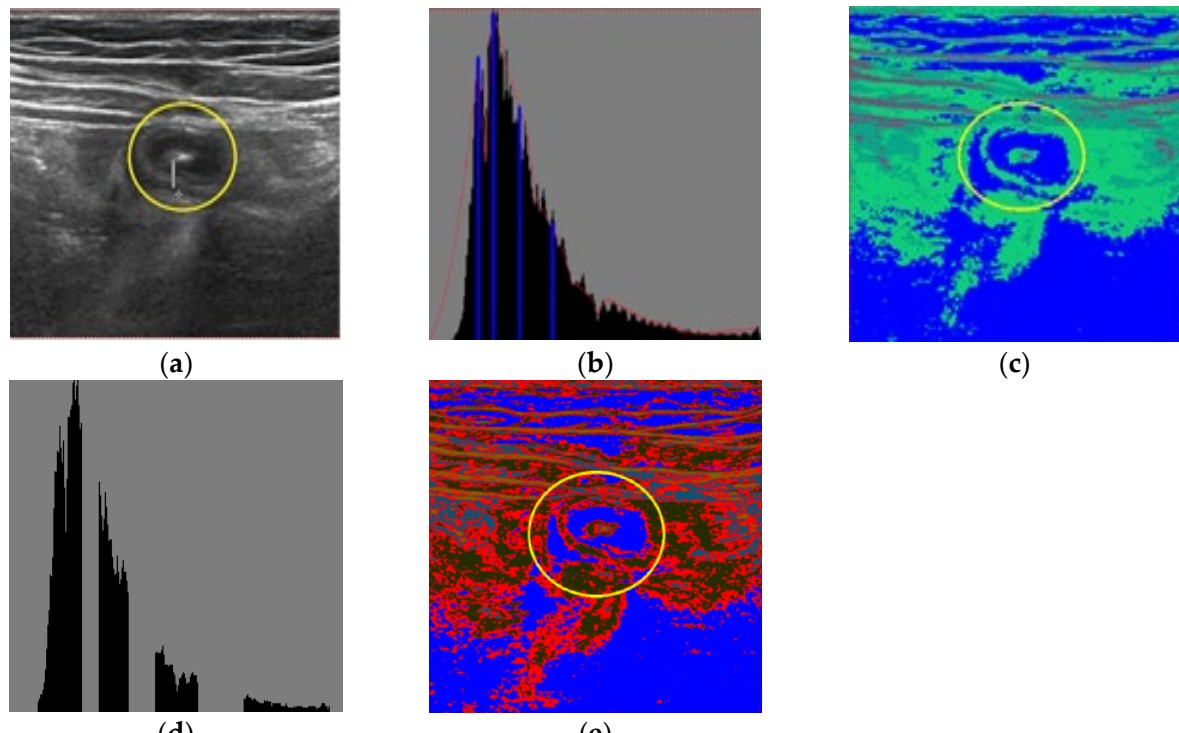

**Figure 7.** Effect of double layered learning structure. (**a**) Input, (**b**) Layer-1 histogram, (**c**) ORFCM result, (**d**) Layer-2 histogram, and (**e**) DORFCM result.

## 4. Conclusions

Abdominal ultrasound examination is the preferred first-hand modality for the diagnosis and management of the patients with clinically suspected appendicitis, especially in emergency and for pediatric or pregnant women patients. However, the sonographic diagnostic criteria for appendicitis, such as diameter, compressibility, wall thickness, and hyperaemia, rely on complete identification or visualization of the appendix. Unfortunately, the sonographic appendix visualization rate can vary significantly with respect to the experience of the examiners. It becomes even more challenging when abdominal ultrasound could not correctly visualize the inflamed appendix.

In this paper, we propose a robust automatic segmentation method for identifying inflamed appendix from abdominal ultrasound images. FCM based pixel clustering approach has shown better performance than edge detection methods, but FCM is sensitive to the existence of outlier thus outlier rejection method is necessary for the robustness. We apply this outlier rejection FCM to the double-layered learning structure in that the first layer clustering result become the input of the second layer outlier rejection FCM with modified learning parameters and cluster center update rule. This double-layered learning mitigates the tendency of local maximum that FCM can fall into. Thus, our proposed DORFCM is a hybrid method to combine outlier-rejection strategy for efficient noise removal and double-layered learning structure for mitigating misguidance of optimization from single ORFCM learning method in inflamed appendix detection.

In experiment with 100 real world inflamed sonographic images, the proposed method (DORFCM) extracts target objects in almost all (98 out of 100) cases and traditional FCM, standard ORFCM and DFCM extracts only 83, 88, and 91 cases, respectively. Furthermore, when we take a human expert's object identification as the ground truth, DORFCM scores better in all criteria, accuracy, precision, and recall, than DFCM and single layered ORFCM significantly in pixel area computation. Thus, DORFCM shows outlier rejection effect and double layered learning effect in our experiment and, thus, we may claim this approach is more robust than standard methods.

**Author Contributions:** Conceptualization, K.B.K. and D.H.S.; methodology, K.B.K.; software, K.B.K. and H.J.P.; analysis, K.B.K. and D.H.S.; resources, K.B.K.; data curation, K.B.K.; writing—original draft preparation, K.B.K., D.H.S. and H.J.P.; writing—review and editing, K.B.K., D.H.S. and H.J.P.; visualization, D.H.S. and H.J.P.; super-vision, K.B.K. and H.J.P.; project administration, K.B.K. All authors have read and agreed to the published version of the manuscript.

**Funding:** This research received no external funding.

**Institutional Review Board Statement:** Not applicable.

**Informed Consent Statement:** Informed consent was obtained from all subjects involved in the study.

**Data Availability Statement:** The data presented in this study are available on request from the corresponding author. The data are not publicly available due to Institutional regulations.

**Conflicts of Interest:** The authors declare no conflict of interest regarding the publication of this paper.

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
