# Peer review of "Robust Automatic Segmentation of Inflamed Appendix from Ultrasonography with Double-Layered Outlier Rejection Fuzzy C-Means Clustering"

_applsci, doi:10.3390/app12115753_

Round 1
Reviewer 1 Report
The authors propose a robust automatic segmentation method for inflamed appendix identification. There is used the fuzzy c-means clustering (FCM) algorithm within a double-layered learning structure for the extraction of the target inflamed appendix area.
The paper is interesting.
The DORFCM method presents a very good successful rate of up to 98%.
Are there also other similar methods that could be studied for an extended comparison (beside FCM and DFCM)?
However, the conclusions must be further extended for a better highlighting of the authors contributions versus other similar topic published papers.
Minor English language improvements can be done.
Reviewer 2 Report
Fuzzy c-means is a mature method that generates good results in image processing and analysis. The methodology presented is a general one and in my opinion I don't think it generates valid results. I think the image analysis process should be detailed with the related evidence. Congratulations on your work, this is an interesting topic and I think it is up to date but needs to be improved.
